# Analysing miRNA-Target Gene Networks in Inflammatory Bowel Disease and Other Complex Diseases Using Transcriptomic Data

**DOI:** 10.3390/genes13020370

**Published:** 2022-02-18

**Authors:** John P. Thomas, Marton Ölbei, Johanne Brooks-Warburton, Tamas Korcsmaros, Dezso Modos

**Affiliations:** 1Organisms and Ecosystem, Earlham Institute, Norwich NR4 7UZ, UK; drjohnpthomas@gmail.com (J.P.T.); marton.olbei@earlham.ac.uk (M.Ö.); dezso.modos@quadram.ac.uk (D.M.); 2Gut Microbes and Health Programme, Quadram Institute Bioscience, Norwich NR4 7UQ, UK; 3Department of Gastroenterology, Norfolk and Norwich University Hospital, Norwich NR4 7UY, UK; 4Department of Gastroenterology, Lister Hospital, Stevenage SG1 4AB, UK; johannebrooks@gmail.com; 5Department of Clinical, Pharmaceutical and Biological Sciences, University of Hertfordshire, Hatfield AL10 9EU, UK; 6Division of Digestive Diseases, Department of Metabolism, Digestion and Reproduction, Imperial College London, London SW7 2BX, UK

**Keywords:** miRNA, network biology, ulcerative colitis, Crohn’s disease, transcriptomics, microarrays, inflammatory bowel disease

## Abstract

Patients with inflammatory bowel disease (IBD) are known to have perturbations in microRNA (miRNA) levels as well as altered miRNA regulation. Although experimental methods have provided initial insights into the functional consequences that may arise due to these changes, researchers are increasingly utilising novel bioinformatics approaches to further dissect the role of miRNAs in IBD. The recent exponential increase in transcriptomics datasets provides an excellent opportunity to further explore the role of miRNAs in IBD pathogenesis. To effectively understand miRNA-target gene interactions from gene expression data, multiple database resources are required, which have become available in recent years. In this technical note, we provide a step-by-step protocol for utilising these state-of-the-art resources, as well as systems biology approaches to understand the role of miRNAs in complex disease pathogenesis. We demonstrate through a case study example how to combine the resulting miRNA-target gene networks with transcriptomics data to find potential disease-specific miRNA regulators and miRNA-target genes in Crohn’s disease. This approach could help to identify miRNAs that may have important disease-modifying effects in IBD and other complex disorders, and facilitate the discovery of novel therapeutic targets.

## 1. Introduction

Inflammatory bowel disease (IBD) is a chronic immune-mediated disease, predominantly affecting the gastrointestinal tract. Although the aetiology of the disease is unclear, it is thought to arise due to complex interactions between multiple genetic risk factors, environmental factors, and the gut microbiota [1,2]. The two major clinical subtypes of IBD include Crohn’s disease (CD) and ulcerative colitis (UC). As with other complex diseases, the majority of genetic risk variants associated with IBD occur within non-coding regions of the genome [3]. These non-coding single nucleotide polymorphisms (SNPs) contribute to disease pathogenesis by affecting the regulation of gene expression. For instance, non-coding SNPs may affect splicing, modify the function of long non-coding RNAs (lncRNAs), alter transcription factor binding sites (TFBS) in promoter regions and introns, and also impact microRNA (miRNA) target sites (miRNA-TS). In recent years, there has been increasing evidence that miRNAs exert important gene regulatory effects and contribute to the pathogenesis of IBD and other complex disorders [4]. Whilst experimental approaches have been helpful for elucidating the role of miRNAs in disease, such methods are laborious, expensive, and time-consuming to perform. Bioinformatics analysis of gene expression data using state-of-the-art database resources and systems biology approaches has the potential to yield powerful predictions for the functional effects of miRNAs in gene regulation. This can facilitate hypothesis-driven validation experiments which can accelerate our understanding of miRNAs in IBD as well as other complex diseases. However, there is a lack of established pipelines or protocols for predicting the gene regulatory effects of miRNAs in complex disease-associated gene networks. In this technical note, we provide a protocol that can be applied to gene expression data for analysing the gene regulatory consequences of miRNAs in complex diseases such as IBD.

### 1.1. A Short Primer on miRNAs

A variety of non-coding RNAs (ncRNAs) are present within a cell. Some exert important housekeeping activities (e.g., ribosomal RNA and transfer RNA), whilst others impart regulatory effects on gene expression. Such regulatory ncRNAs include short ncRNAs (<200 nucleotides in length) and long ncRNAs (>200 nucleotides in length). miRNAs, small interfering RNAs (siRNAs), and piwi-interacting RNAs (piRNAs) comprise the three main types of regulatory short ncRNAs [5]. Here, we focus our attention on miRNAs.

miRNAs are single-stranded RNA molecules that are, on average, 22 nucleotides in length. miRNAs are generated from double-stranded RNA hairpin precursors (the primary miRNA or pri-miRNA) through a multi-step maturation process (Figure 1) [6]. These miRNA precursors are typically found in clusters, most frequently within intronic regions of protein-coding genes and intergenic regions of the genome [7]. The pri-miRNA is often more than 1000 nucleotides in length. It contains a 60–120 nucleotide RNA hairpin which is cleaved by the enzyme Drosha in the nucleus of the cell, to generate the precursor-miRNA (pre-miRNA). Pre-miRNAs are then exported to the cytoplasm via the exportin-5 protein, where they are cleaved by the enzyme Dicer to form mature miRNA [8]. Mature miRNA binds to Argonaute proteins to form the RNA-induced silencing complex (RISC) which functions to prevent the translation of target messenger RNA (mRNA) [9]. The miRNA in the RISC binds to a 13–16 nucleotide sequence in the 3’ untranslated region (UTR) region of one or more target mRNAs. In animal cells, only the 2nd–8th nucleotides (i.e., the seed sequence) of the mRNA have perfect complementarity to the miRNA [10]. The binding of the miRNA-RISC to the mRNA causes the inhibition of translation and also accelerates the deadenylation of the mRNA polyA tail, resulting in earlier degradation of the mRNA [11].

Thus, miRNAs are able to fine-tune gene expression at the post-transcriptional level. One miRNA can potentially interact with several mRNAs and the same mRNA may be targeted by many different miRNAs. In this way, miRNAs and mRNAs form complex networks of gene regulation [12]. Perturbation of these miRNA-mRNA networks can lead to cellular dysfunction and ultimately pathological states. Indeed, the disruption of miRNA-gene regulatory networks has been shown to contribute towards the pathogenesis of cancers, neurological disorders, cardiovascular diseases, and chronic immune-mediated disorders such as IBD [13].

### 1.2. miRNAs in IBD

Over the past decade, several studies have revealed that levels of certain miRNAs are altered in IBD patients in comparison to healthy individuals, such as miR-124, miR-320, miR-21, miR-31, and miR-141 (reviewed in detail in [14]). Recently, it was demonstrated that disease stage may also influence miRNA levels in the same disease phenotype. Verstockt et al. showed that CD patients at different stages of their disease (i.e., newly diagnosed CD, late-stage CD, and post-operative recurrent CD) have differences in gene and miRNA expression profiles [15]. The investigators found enhanced dysregulation of miRNA and gene expression networks in ileal biopsies from newly diagnosed CD and post-operative CD, in comparison to late-stage CD, suggesting that miRNA dysregulation may play a key role at the mucosal level in early-stage CD, and also after “resetting” the disease through surgery. In addition, others have demonstrated that miRNAs could be potential biomarkers in IBD for therapeutic response as well as diagnosis. Viennois et al. found a peripheral blood miRNA signature from mice which was able to predict the response to anti-TNF therapy in colitic mice [16]. Importantly, they demonstrated that this signature could identify UC patients with 83.3% accuracy. Similarly, Wu et al. identified a miRNA signature from blood which could be utilised to distinguish active UC and active CD from healthy controls [17]. These studies suggest that miRNAs may play an important role in IBD pathogenesis.

The functional effects of miRNAs in IBD have been investigated through murine models. Knockout of miR-21 in the dextran sodium sulphate (DSS)-induced fatal colitis model resulted in reduced levels of inflammation and improved survival in mice [18]. miR-31 has been shown to directly target IL-25 by binding to its mRNA at the 3’ UTR [19]. In addition, altered levels of miR-31 can shape IL-12/23-mediated Th1/Th17 pathways in the murine colon and ameliorate 2,4,6-trinitrobenzene sulfonic acid (TNBS)-induced colitis [19]. miR-141 has been found to target CXCL12β, with the downregulation of miR-141 resulting in enhanced CXCL12β release and CXCL12β-mediated leukocyte migration in colitic mice. Thus, miRNAs may contribute to IBD pathogenesis through a variety of pathogenic mechanisms.

## 2. Bioinformatics Approaches for Predicting miRNA Function in Complex Disease

As experimental approaches can be technically challenging and resource-intensive, researchers have been increasingly employing computational approaches for predicting the effect of miRNAs in the pathogenesis of complex disorders. The following section highlights a few examples on how bioinformatics approaches have been utilised to predict miRNA-disease associations. These can be broadly divided into similarity-based and transcription-based approaches.

### 2.1. Similarity-Based Methodologies

There has been a substantial rise in the application of machine learning approaches in biology. Similarity-based miRNA-disease association is one such example. The fundamental premise of this approach is that miRNAs involved in certain diseases are also likely to be involved in similar diseases based on the miRNA target profile or the disease pathomechanism. Most of these methods employ various machine learning and random walk approaches.

An example of this method is the study by Sumathipala and Weiss [20]. In their work, the authors used network diffusion approaches on multi-omics biological data in their miRNA-disease association prediction (MAP) pipeline to predict and prioritise miRNA-disease associations without the use of a priori miRNA-disease data [20]. To do this, they combined genomic and transcriptomic datasets into a miRNA-gene-disease tripartite network and applied a network diffusion algorithm to rank miRNA candidates for a disease. MAP was able to accurately predict miRNA-disease associations for four cancer types including lung cancer, renal cancer, lymphoma, and breast cancer. This pipeline could potentially be applied to IBD and other complex disorders too.

Mørk et al. developed the miRPD tool, which infers miRNA-protein-disease associations using a similarity-based approach [21]. The workflow uses network analysis on currently known or predicted miRNA-protein associations and text-mined protein-disease associations, ranked by confidence. In their case study, they identified a statistically significant relationship between miR-181 and diabetes mellitus. Using the available protein interaction information, they were able to highlight the potential importance of miR-181 on glutamate decarboxylase 2—a key protein implicated in type I diabetes. The analysis also revealed a strong association between miR-181 and sirtuin-1. This corresponds to previous experimental work which has revealed that the downregulation of miR-181a levels can lead to sirtuin-1 upregulation and increased insulin sensitivity in hepatic cells [22].

Another interesting similarity-based method is the ranking-based k-nearest neighbour for miRNA-disease association prediction (RKNNMDA) tool developed by Chen et al. [23]. The approach uses miRNA functional similarity, disease semantic similarity, and already known interaction profile similarity derived from known miRNA-disease associations to identify the k-nearest neighbours for both the disease and miRNAs. The resulting neighbours are re-ranked using a support vector machine ranking model following which weighted voting identifies the final miRNA-disease associations. In three case studies (colon, oesophageal, and prostate cancer) the method captured more than 80% of the top 50 miRNAs. As the method does not use any such *a priori* data, it is a suitable tool to discover miRNAs for diseases that do not yet have any established miRNA associations.

### 2.2. Transcription-Based Methodologies

Transcription-based methodologies utilise transcriptomics data to derive miRNA-disease predictions. This requires at least an mRNA transcriptome, but ideally the miRNA transcriptome should be available. Transcriptome-based approaches are the most common way for analysing the role of miRNAs in diseases. Xu et al. demonstrated the utility of such an approach [24]. They developed a novel method that could prioritise disease-specific miRNAs, using matched miRNA and mRNA expression data from 11 cancer types. Their prediction model uses miRNA–mRNA interactions collected from multiple databases and is based on the assumption that diverse diseases with phenotype associations show similar molecular mechanisms. The systematic prioritisation of disease-specific miRNAs is completed by using known disease genes and context-dependent miRNA-target interactions derived from matched miRNA and mRNA expression data, independent of known-disease miRNAs. For IBD, a similar methodology was used by Verstockt and colleagues as previously mentioned [15].

With the exponential increase in the availability of disease-specific gene expression data, especially in complex diseases such as IBD [25], there are new opportunities to further investigate the role of miRNAs in disease pathogenesis. In the following section, we provide a technical summary describing the steps required to perform transcriptome-based miRNA-target gene analysis in complex disorders.

## 3. Materials and Methods: Protocol for Transcriptome-Based miRNA-Target Gene Analysis for Complex Disorders

The protocol is summarised in Figure 2 (the various steps in Figure 2 correspond to the parentheses in the section below).

**Selecting an experimentally validated miRNA–mRNA target database**: For precise interactions, the first step is to choose databases containing validated interaction data such as Tarbase [26] or miRTarBase [27]. These interactions have high confidence.**Strengthening predictions using additional miRNA–mRNA target databases**: Manually curated databases do not have all miRNA-target gene interactions. Hence, it is useful to add an additional miRNA-target gene interaction database which contains predicted miRNA-target gene interactions. The best approach is to use multiple data sources from at least two complementary methods, e.g., TargetScan and miranda/mirSVR.**Combining the miRNA-target gene networks using the same miRNA IDs**: Unfortunately, miRNAs are not always consistently named between databases. miRBase is the de facto database for miRNA families and sequences and contains a basic ID mapping tool [28,29]. MiRBase has an R package for application programming interface (API) access called miRBaseConverter [30]. Please note, in the case of TargetScan, the miRNA family-based conversions can cause many-to-many mapping issues.**Combining the miRNA-target gene networks using the same target gene IDs**: The same problem can arise when mapping target genes. A good solution is to use the biomaRt R package, or download the various annotation tables from UniProt [31] or Ensembl biomart [32]. It is important to choose one type of gene ID when constructing the gene network.**Preparing gene expression data**: The transcriptomics data can be downloaded with normalised log2 counts or expression values from Gene Expression Omnibus (GEO) [33], ArrayExpress [34], or analysed in-house. Ideally, both mRNA and miRNA transcriptome data are available, but often this is not the case. In the case of RNA-seq data analysis, we suggest using the normalised count table with genes considered as expressed if they have more than 1 count per million. However, the exact normalisation between the samples and the filtering of weakly expressed genes are choices the researcher should make depending on the sequencing depth, read quality, and read length [35,36].**Finding differentially expressed genes**: From the normalised counts, the next step is to calculate differentially expressed genes (DEGs), e.g., using the limma package from R. Limma uses the moderated t-statistic to calculate differential gene expression [37]. We recommend |log2FC| > 1 and Benjamini–Hochberg (BH) corrected *p* value < 0.05, but again it can change depending on the researcher’s questions and quality of samples. It is important that these results are corrected for false discovery rate and filtered by fold change as well.**Enrichment of potential miRNA targets in the differentially expressed genes (if miRNA expression values are not available)**: If the miRNA expression values are not available, it is possible to enrich potential regulatory miRNAs using the constructed miRNA gene network. To do this, the used gene network needs to be transformed as a gene matrix (GMT) file. The constructed GMT file can be used as an input of a gene set enrichment analysis (GSEA) [38]. In this case, the expression values are used for calculations. Moreover, the DEGs with the GMT file can be used for enrichment analysis, for instance using gprofiler2 [39]. It is important that the enrichment background, the intersection of miRNA-regulated genes, and the measured transcripts are utilised.**Building an anticorrelation network (if the miRNA expression data are available)**: If miRNA expression data are available, then the prediction of miRNA and target genes can be calculated, e.g., from the TaLasso algorithm [40] which has an R package available or the miRLAB package [41]. The correlation-based network needs to be filtered with the previously generated miRNA-target gene network to identify the dataset-specific miRNA-target gene interactions. The MAGIA2 method can be used for steps 2–8 to produce the output networks [42]. We suggest using high correlation values such as the 25th percentile of the most negatively correlating miRNAs.**Visualising the networks**: After creating the networks where the potential miRNA targets are given, the next step is to visualise the network using network visualisation tools. The most commonly used visualisation tool in network biology is Cytoscape [43]. If programmatic access is required, the R and python software package igraph can be used [44].**Analysing the resulting visualised network**: The most common way to analyse the visualised network is by looking at the hubs in the network, i.e., nodes that have a high number of edges or degree. Those target genes or miRNAs with a high degree (“hub” genes/miRNAs) have a greater chance to be involved in the disease of interest (in this case IBD). For identifying these nodes, the Cytoscape network analyser can be used. We suggest that the top ten percent of highest degree nodes can be considered as hub genes, but this also depends on the degree distribution of the network.**Functional analysis**: The next step is to evaluate what functions of the network are potentially regulated by miRNAs compared to the whole network. This can be performed by gene ontology enrichment analysis, e.g., the aforementioned gprofiler2 [37] package or the GOrilla analysis tool [45]. An important caveat to bear in mind is that miRNA target genes are limited by the used miRNA-target gene network. Therefore, the background of any enrichment analysis needs to be modified specifically. In the background, only the genes in the miRNA-target gene network should be utilised. If the differentially expressed genes are directly used for building the miRNA-target gene network, then it is advisable not to perform enrichment analysis because this would, in effect, simply test differentially expressed genes again.**Experimental validation**: The next step would be the experimental validation of the detected miRNA and target gene anti-correlation, which is beyond the scope of this technical note.

The R codes for this workflow are available on the project GitHub page: https://github.com/korcsmarosgroup/miRNA_target_gene_workflow.

## 4. Results—Case Study in Crohn’s Disease

We ran the above workflow on transcriptomics data from the aforementioned study by Verstockt et al. [15]. In their study, the authors extracted and examined gene and miRNA expression data from ileal CD patient biopsies at various stages of their disease, as well as healthy controls. We used the TargetScan database as a prediction-based miRNA-target gene interaction database, and the mirTarBase database as the source of validated miRNA-target gene interactions. TargetScan has two complementary prediction methods, but each of them rely on the researcher selecting the most suitable cut-off for the analysis. The TargetScan Probability of Conserved Targeting (PCT) score measures the degree of conservation for each miRNA target site, while the Context+ score is related to the strength of the miRNA-target interaction. We used PCT > 0.5 and Context+ score% > 95% as cut-offs, and after mapping we used the intersecting set of these two miRNA-target gene connections. This is a really stringent, low coverage result. We followed the left side of the workflow in Figure 2 in this case study, calculating the DEGs, and finding the most regulating miRNAs and regulated genes (Figure 3a–c). Following this, we used the correlation methodology relying on relevant functions from the miRLAB R package [41]. The correlated networks were filtered with the above-mentioned TargetScan network-miTarBase union network. On this side of the workflow, selecting the most appropriate correlation cut-off is a challenging question. In this case, we used a stringent correlation cut-off of −0.75 between miRNA and target gene expression after plotting the distribution of correlation scores. The networks of both CD and control cases were calculated and the differences computed in terms of interactions and enrichment in relevant functions. The enriched functions specific to CD-regulated genes can be found in Figure 3d–f. Please note that the networks themselves in this case were not informative en bloc, as they were large hairballs which made it difficult to identify the most important interactions. To combat this, we visualised only the interactions where both the miRNA and the target gene were specific to the controls and CD patients.

The protocol which we have presented here answers different questions compared to the study by Verstockt et al. The presented network (Figure 3a) reveals the miRNAs which regulate most of the DEGs in early-onset CD. These miRNAs are not necessarily differentially expressed in the network. One of the most regulating miRNAs was mir-141, which is a known miRNA implicated in IBD [14]. In addition, this methodological approach could suggest novel miRNA-target gene interactions which are involved in CD pathogenesis. Furthermore, the resulting correlation network highlighted differences in biological processes influenced by miRNA-target gene networks between healthy and CD states. For instance, cellular response to stress mesenchyme development and various cancer pathways were specifically enriched in CD patients (Figure 3d–f). Our analysis in Figure 3g,h emphasised connections where both the miRNA and the target gene are specific for a certain disease state (Figure 3g,h). Such interactions and the weight of anti-correlation suggests there may be a CD-specific interaction between the miR-450b and CD4 T cell co-receptor.

## 5. Discussion

There are several important considerations to be made when implementing the above protocol. The most important is choosing the appropriate miRNA-target gene databases (Table 1). The best approach is to use multiple miRNA-target gene databases to increase the coverage—ideally those that use different types of data. If the aim is to consider only the most plausible interactions, then using interactions which are present in more than one database is desirable.

These various databases are constructed from different source data. The databases with the highest confidence are manually curated databases such as TarBase or miRTarBase. These databases contain miRNA-target connections which have been detected experimentally by miRNA overexpression and mRNA sequencing, RISC immunoprecipitation, and then sequencing and pull-down of miRNA by a tagged miRNA nucleotide (Figure 4a). These methods have been recently reviewed by Li and Zhang [60]. Conservation-based methods (Figure 4b) are an alternative approach used by databases including TargetScan [61] and the older PicTAR [51]. These methods are based on the underlying principle that miRNA binding sites are expected to be conserved between species because they are under evolutionary selection pressure. Due to this, many bioinformatics methods are using sequence homology comparison to predict miRNA-target interactions. These methods are sensitive to the used species and the scoring function of the homology. Each target nucleotide has a different scoring weight in these methods. Biochemistry or free energy-based methods have also been utilised, such as in the Miranda database (Figure 4c). miRNAs often bind to their target sequences without strictly following the Watson–Crick base pair rules. The resultant wobble base pairing can be estimated with hidden Markov model-based free energy quantification. The greatest advantage of such methods is that it is possible to use different sequences to predict them with relative ease. The disadvantage with this method, however, is that the miRNA-target site prediction is based on the chosen free energy cut-off, which can be arbitrary. The fourth major approach are the anticorrelation-based methods (Figure 4d). These methods require both miRNA and mRNA gene expression and they use additional methods to decrease the number of potential miRNA-target interactions. Here, the basic assumption is that if the amount of miRNA increases, the target mRNA level will decrease. As a linear model, this phenomenon can be described using the following equation (Equation (1)) [40]:(1)$$x_{j}=\overset{K}{\underset{k=1}{\sum }}\beta_{jk}\cdot z_{k}\cdot c_{jk}+x_{j}^{0}+\epsilon_{j}$$
where *x_j_* is the expression of gene *j*, *β_jk_* is the linear model’s parameter which tells the effect of miRNA *k* on the target gene *j*, *z_k_* is the miRNA *k* concentration, *c_jk_* indicates whether there is a connection between the miRNA and the target gene, $x_{j}^{0}$ is the original concentration of the gene, and *e_j_* is an error term. This results in a linear model for predicting the parameter *β_jk_*. To decrease the search space and make the model more plausible, *c_jk_* can be 0 or 1 depending on whether the miRNA-target gene interaction is found within other databases. An example is the TaLasso algorithm [40], which finds the solution of the linear model with LASSO regression. Other methods use different ways to predict the effect between a miRNA and a target gene—the *β_jk_* in Equation (1). The Genmir++ algorithm uses a Bayesian network for prediction [62]. The biggest problem of such methods is that it requires both the miRNA and the mRNA transcript levels, which is not always available.

The protocol described in this technical note can be used as a standard approach for understanding the relationship between gene expression and miRNA. However, the protocol requires experimental transcriptomic data and existing miRNA-target information, and assumes the anti-correlation between miRNA and target gene transcript. For discovery of novel miRNAs involved in the pathogenesis of a specific disease, machine learning and similarity-based methods may be more suitable. The problem with this approach is that the predictions are based on similarity and the exact mechanism of miRNA regulation is not necessarily discerned. However, with transcriptome-based methods such as the protocol we have described and the case study example, the resulting network can reveal disease-specific interactions which can shed novel mechanistic insights into disease pathogenesis. It is important to note that both methods are dependent on the quality of the input dataset. In the case of machine learning similarity-based approaches, the input miRNA-disease network contributes to the largest bias of the results. For transcriptome-based analysis, the results depend on the miRNA-target gene networks. Another point to consider is that miRNA target gene information is not always available for every 3p and 5p miRNAs. Moreover, microarray technologies (such as those used in the case study) can have cross-hybridisation effects. In the presented protocol we considered the available predicted miRNA-target gene interactions for the left side of the workflow, whilst for the right side of the workflow we kept the annotation of the microarray chip relevant. Another important consideration for this protocol is that input transcriptomic data from disease-relevant tissues is likely to yield more accurate predictions, as miRNA expression varies widely between tissues [63] and also contributes to the tissue-specific expression of mRNAs in humans [64,65].

## 6. Conclusions

miRNAs play an important role in the regulation of gene expression and contribute to the pathogenesis of complex diseases such as IBD. With the exponential increase in publicly available transcriptomics datasets, there is great potential to leverage this data for advancing our understanding of the role of miRNAs in IBD. In this technical note, we have provided a step-by-step protocol that utilises multiple database resources and systems biology methods to predict the effect(s) of miRNAs in IBD (and other complex diseases) from gene expression data. Using this protocol, we highlight differences between miRNA-target gene networks between CD patients and healthy individuals, which could enable the identification of miRNAs that have important disease-modifying effects and act as novel therapeutic targets [66].

## Figures and Tables

**Figure 1 genes-13-00370-f001:**
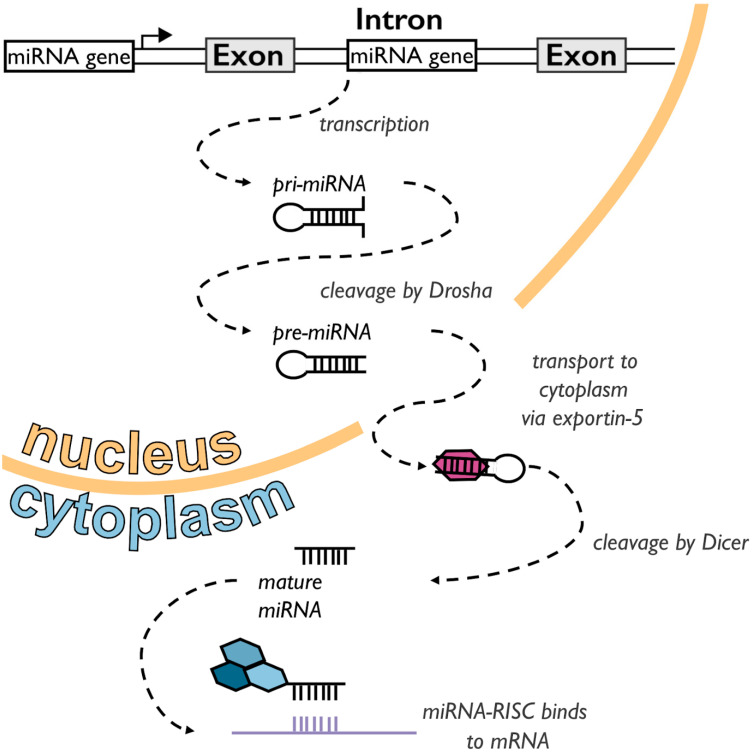
Simplified overview of miRNA synthesis: miRNAs are transcribed from untranslated regions (UTRs) or intronic regions of the genome. The transcribed pri-miRNA is cleaved by the enzyme Drosha into the pre-miRNA form, which is then transported through the nuclear pore complex to the cytoplasm via the enzyme exportin-5. The enzyme Dicer cleaves the pre-miRNA into its mature form, where it can start regulating mRNA following binding to the RISC complex.

**Figure 2 genes-13-00370-f002:**
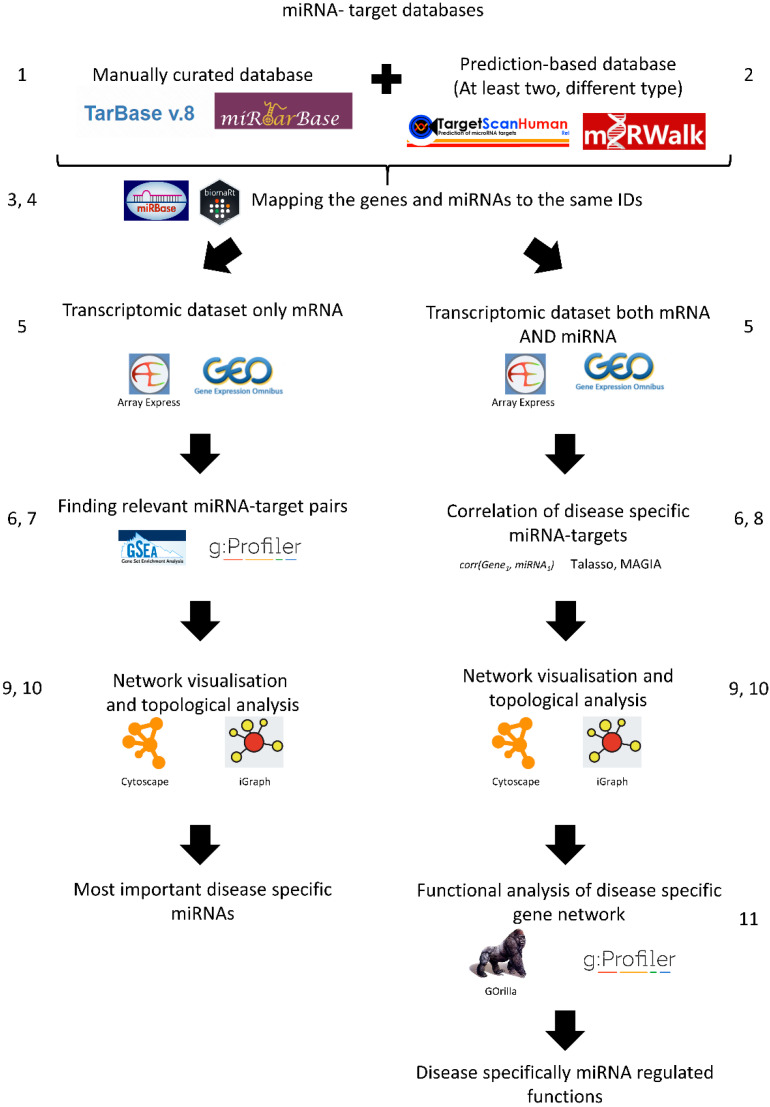
Analysing miRNA-target gene networks using gene expression data: a step-by-step protocol.

**Figure 3 genes-13-00370-f003:**
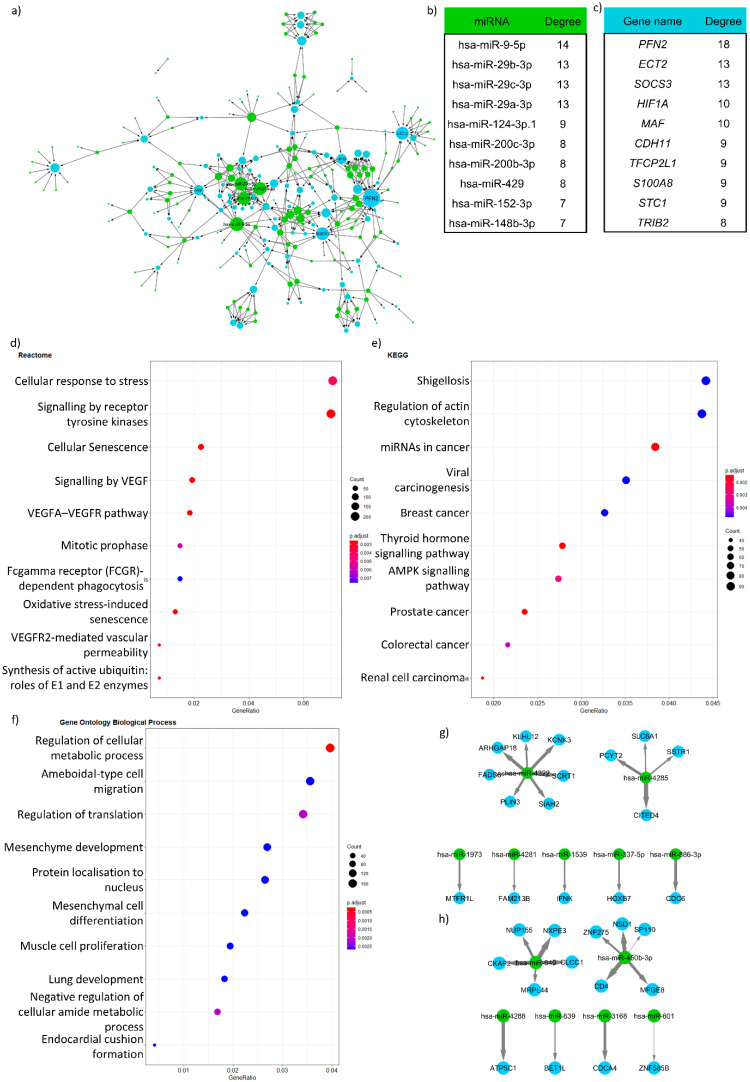
(**a**) miRNA-target gene network using the two types of TargetScan network sources. (**b**) miRNAs regulating the most target genes (**c**) Most regulated differentially expressed genes (**d**–**f**) Overrepresentation analysis using various ontology sources (Reactome, KEGG, and Gene Ontology biological process) of the CD-specific anti-correlation-based miRNA-target gene network. (**g**,**h**) Specific miRNA-target gene interactions for healthy controls (**g**) and CD patients (**h**). The weight of the edges corresponds with the anticorrelation.

**Figure 4 genes-13-00370-f004:**
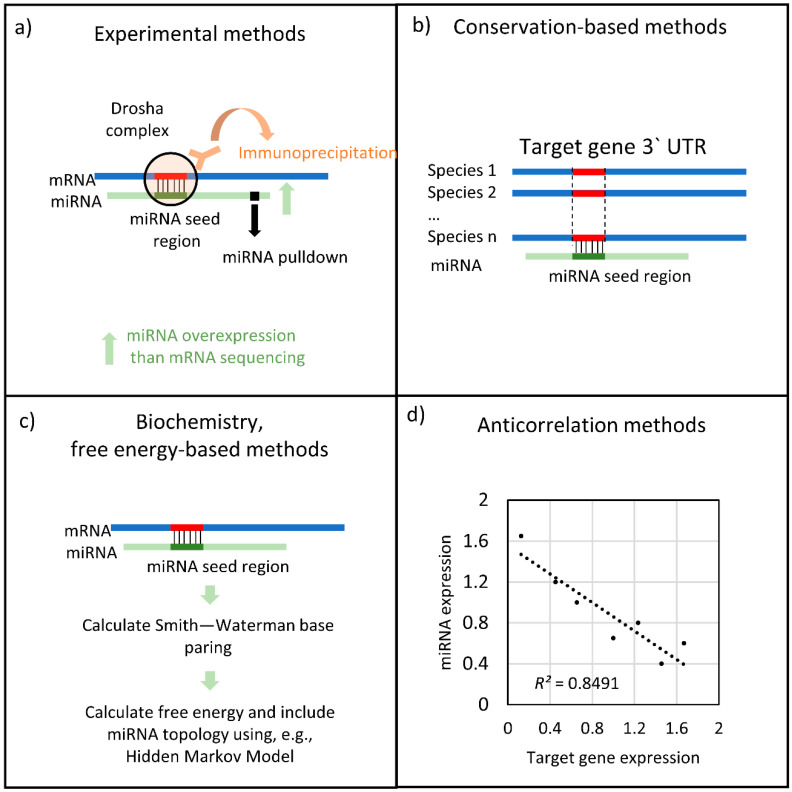
Summary of the various types of miRNA target detection methods.

**Table 1 genes-13-00370-t001:** Various miRNA—target gene databases and tools.

Type	Name	Description	Website (Access Date)	Reference
General	Mirbase	The de facto miRNA central repository for miRNA families and sequences	http://www.mirbase.org (16 January 2022)	[28]
Literature curation	miRTarBase	Experimentally proven miRNA-target gene connections	http://mirtarbase.mbc.nctu.edu.cn (16 January 2022)	[46,47]
Literature curation	TarBase	Experimentally validated miRNA-target gene connections	http://carolina.imis.athena-innovation.gr/diana_tools/web/index.php?r=tarbasev8%2Findex (16 January 2022)	[26]
Literature curation	miRDeathDB	Small database of experimentally validated miRNA-target interactions related to cell death	http://www.rna-world.org/mirdeathdb (25 April 2016)	[48]
Literature curation	miR2Disease	Small literature curation database for miRNAs associated with diseases	http://www.mir2disease.org (16 January 2022)	[49]
Conservation based	Targetscan	The largest miRNA-target gene prediction database based on sequence homology	http://www.targetscan.org/ (16 January 2022)	[50]
Conservation based	PicTar	Old sequence homology-based miRNA-target database	http://pictar.mdc-berlin.de (16 January 2022)	[51]
Biochemistry based	Miranda	A free energy-based algorithm for miRNA-target prediction currently unavailable, but still widely used	http://www.microrna.org/microrna/home.do (16 June 2020)	[52]
Biochemistry and conservation based	SVMicrO	Two-stage support vector machine-based miRNA-target prediction algorithm and database integrating biochemistry, alignment, and conversation features of the target site and the miRNA	http://compgenomics.utsa.edu/svmicro.html (16 January 2022)	[53]
Biochemistry based	PITA	Thermodynamics-based prediction tool which incorporates the target’s accessibility	https://genie.weizmann.ac.il/pubs/mir07/mir07_prediction.html (16 January 2022)	[54]
Expression based	hoctar	It uses various prediction tools and then multiple miRNA and target gene expression datasets to calculate the potential miRNA-target gene connections	https://hoctar.tigem.it/ (16 January 2022)	[55]
Expression based	CAPE RNA	miRNA-target gene prediction tools using discrete mRNA and miRNA levels (middle, high, low)	https://sourceforge.net/projects/caperna (16 January 2022)	[56]
Literature curation and predicted targets	miRecords	Joint effort of multiple prediction tools and validated targets. Last updated in 2013.	http://c1.accurascience.com/miRecords (16 January 2022)	[57]
Large collection of multiple different resources	mirwalk	Contains 13 different target prediction methods and it generates predictions for the whole length of the genes	http://zmf.umm.uni-heidelberg.de/apps/zmf/mirwalk2 (16 January 2022)	[58]
Large integrated database	miRabel	Contains integrated predictions from multiple prediction databases and also the experimentally validated informations of miRecords and miRTarBase	http://bioinfo.univ-rouen.fr/mirabel (16 January 2022)	[59]
Integrated tool for expression-based prediction	MAGIA	Prediction tool using miRNA and mRNA gene expression to integrate with various miRNA-target prediction algorithms	http://gencomp.bio.unipd.it/magia2/start (25 August 2018)	[42]

## Data Availability

The workflow code is available at https://github.com/korcsmarosgroup/miRNA_target_gene_workflow.

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
