# Peer review of "Analysing miRNA-Target Gene Networks in Inflammatory Bowel Disease and Other Complex Diseases Using Transcriptomic Data"

_genes, 2022, doi:10.3390/genes13020370_

Round 1

Reviewer 1 Report

The paper is devoted to an actual issue for all researchers working with omics data. The authors suggest a protocol for the analysis of miRNA-mRNA networks helping to elucidate the role of miRNA in complex diseases. However, the paper seems to be rather a review of the protocol than the analysis of “miRNA-Target Gene Networks in IBD” using the proposed protocol that I expected based on the paper title and abstract. The paper needs major revision.

Minor comments:

I think that it is not necessarily writing about “what is miRNAs” in the Abstract. More important is to note the actuality of miRNA study in IBD.

 “..perturbations in miRNAs..” (Abstract) – what do you mean? Alterations in miRNA expression/miRNA-mRNA regulation disruptions/alterations in miRNA maturation etc. Please, clarify.

Part 1.1. A short primer on miRNAs - Please, clarify differences between double-stranded RNA hairpin precursors (pri-miRNA) and precursors miRNA (pre-miRNA).

Figure 1 – The term “UTR” is an abbreviation for the untranslated region on mRNA.

Major comments:

miRNA database, as well as miRNA-Seq output data, cannot define some 3p and 5p miRNAs. At the same time, 3p and 5p miRNAs matured from the same precursor can have different effects. The authors should describe what to do in this case.

The major note for the paper is that the authors did not represent the analysis of real transcriptomic data (e.g. from public databases) on IBD using their algorithm with a subsequent discussion of its advantages and disadvantages, as well as biological and/or practical results (miRNA-RNA regulatory networks). This paper is now only the protocol and its review without real data analysis.

Author Response

We thank the Reviewer for the careful reading of our Technical note, and for the useful comments that helped us to improve the manuscript.

Minor comments:

I think that it is not necessarily writing about “what is miRNAs” in the Abstract. More important is to note the actuality of miRNA study in IBD.

Thank you for your comment. We have now addressed this in the Abstract. 

 “..perturbations in miRNAs..” (Abstract) – what do you mean? Alterations in miRNA expression/miRNA-mRNA regulation disruptions/alterations in miRNA maturation etc. Please, clarify.

Thank you - we have clarified this in the Abstract.

Part 1.1. A short primer on miRNAs - Please, clarify differences between double-stranded RNA hairpin precursors (pri-miRNA) and precursors miRNA (pre-miRNA).

We have added a line to clarify the difference between pri-miRNA and pre-miRNA. 

 Figure 1 – The term “UTR” is an abbreviation for the untranslated region on mRNA.

We have now added the term.

Major comments:

miRNA database, as well as miRNA-Seq output data, cannot define some 3p and 5p miRNAs. At the same time, 3p and 5p miRNAs matured from the same precursor can have different effects. The authors should describe what to do in this case. 

Thank you for raising this point. We added an extra sentence regarding this in the discussion. Sadly It is not an easily addressable problem in the field, especially with legacy data. 

The major note for the paper is that the authors did not represent the analysis of real transcriptomic data (e.g. from public databases) on IBD using their algorithm with a subsequent discussion of its advantages and disadvantages, as well as biological and/or practical results (miRNA-RNA regulatory networks). This paper is now only the protocol and its review without real data analysis.

Thank you for this great suggestion and feedback. Accordingly, we added a case study of Crohn's disease data analysis and provided code in a Github repo ready for analysis. We also note that this paper is meant to be a protocol paper, encouraged by the Editorial Office. We fully agree with the Reviewer that indeed a real data demonstration was missing from our manuscript that has now been added to the revision.

Reviewer 2 Report

The report is a technical note that provides a template to analyze miRNA-Target gene networks using transcriptomic data. The study is comprehensive and presents a systematic way to analyze miRNA/mRNA data. A summary of the software is provided which is very useful. There are 2 minor issues. It would have been useful if a practical example is provided with IBD as a focus. Another issue is how tissue-specific expression profiles could be used to improve specificity.

Author Response

Thank you for the positive feedback on our work and for these key comments. In the revision, we discuss now that tissue-specific expression profiles could improve the specificity of the analysis. We have also undertaken a practical example using Crohn’s disease data as a case study, and presented it in detail in the revised manuscript.

Round 2

Reviewer 1 Report

The corrected manuscript was not marked up using the “Track Changes” function which made it difficult to revise.

Materials and Methods section

Step 5. Authors should tell about what is the CPM (count per million) threshold for microRNAs/transcripts for further analysis. Note, expression of microRNA/transcripts with low CPM is difficult or impossible for further experimentally validation.

Step 6. Add the recommended threshold for DEGs for further enrichment.

Step 8. Add the recommended correlation value for miRNA-mRNA pairs indicating their potential disease-specific interactions.

In general, my opinion is that the proposed protocol is suitable for the study of various diseases and, moreover, is often used in this way. In the protocol itself, I do not find any features that should be applied specifically for the IBD. The article can be retitled as “Approach for miRNA-Target Gene Networks analysis in Complex Diseases” and in Abstract to note, that the protocol was validated on Bowel Disease (it is only a recommendation).

Author Response

The corrected manuscript was not marked up using the “Track Changes” function which made it difficult to revise.

We are sorry that the uploaded tracked-changes version was not available for you to review. We have uploaded the recent revision again with track changes and pointed out the importance of this file for the editorial office, so we hope this time, it will be shared with you.

Materials and Methods section

Step 5. Authors should tell about what is the CPM (count per million) threshold for microRNAs/transcripts for further analysis. Note, expression of microRNA/transcripts with low CPM is difficult or impossible for further experimentally validation.

Thank you for this comment. It is really difficult to choose a count per million threshold in RNA-seq analysis. In the literature, the most commonly used CPM threshold is 1, to the best of our knowledge.  We added discussion points and references pointing to further parameters in consideration into the manuscript. 

Step 6. Add the recommended threshold for DEGs for further enrichment.

Thank you for pointing this out, we have added the threshold to the revised manuscript.

Step 8. Add the recommended correlation value for miRNA-mRNA pairs indicating their potential disease-specific interactions.

Thank you for pointing this out, we have added the threshold to the revised manuscript. 

In general, my opinion is that the proposed protocol is suitable for the study of various diseases and, moreover, is often used in this way. In the protocol itself, I do not find any features that should be applied specifically for the IBD. The article can be retitled as “Approach for miRNA-Target Gene Networks analysis in Complex Diseases” and in Abstract to note, that the protocol was validated on Bowel Disease (it is only a recommendation)."

We are grateful for the reviewer’s suggestion, however, we submitted our protocol to an IBD-focused special issue of Genes. Nonetheless, we added a sentence in the abstract and mentioned in the Conclusions section that the protocol can be used in other complex diseases.
